# OpenReview forum: "When "Competency" in Reasoning Opens the Door to Vulnerability: Jailbreaking LLMs via Novel Ciphers"
_NeurIPS.cc/2025/Workshop/Reliable_ML — NeurIPS 2025 - Reliable ML Workshop_

### Official Review · Reviewer_d1gu · 2025-09-11
**Empirical evidence that uncommon cipher combinations jailbreak LLMs**

**Rating:** 7
**Confidence:** 3

**Review:**

**Summary:**
The paper proposes that using combinations of uncommon ciphers can lead to effective jailbreaking attacks against LLMs. A claim which they support empirically.

**Strengths:**
1) Clarity: The paper is well written and the ideas clear.
2) Evaluation: It seems that the attack-success-rate is comparable other works.

**Weaknesses:**
 Lack of Qualitative Comparisons: I do not see any reference or comparison with what attack rate other techniques lead to. Other techniques are referenced but not a quantitative comparison. However, it seems that the attack success rate is sufficiently high.

**Questions:**
I do not know if I agree with this claim:
as LLMs become more adept at deciphering complex user ciphers many of which cannot be preemptively included in safety training they become increasingly exploitable.

In my opinion this sentence is more complex and not proved by the experiments of the paper.


One attempt to prove is to at least calculate the statistics similar to:
$\mathrm{Pr}_{\text{model } x, \text{cipher } c \text{of complexity} >t}[x \text{ is jailbreaked} | x \text{ solves } c]$. Also it seems that these statistics are decreasing in the experiments of the paper (because LLMs with better reasoning skills have better safety training) and thus disproves the statements.

Can the authors help me understand this?

---

### Official Review · Reviewer_urNq · 2025-09-19

**Rating:** 7
**Confidence:** 3

**Review:**

This paper introduces evidence that current frontier models are vulnerable to novel ciphers, and that improved reasoning capabilities increases their vulnerability. It introduces a benchmark for measuring model capability on the general task of cyphers and a collection of novel ciphers called ACE and LACE. The most convincing evidence comes from evaluation results on GPT-4o where they show known ciphers and novel ciphers with similar decryption rates by the model, but the novel ciphers have a higher ASR. Some of the other results are more mixed however. The GPT-5 results show a higher decryption rate when two ciphers are combine than when either of the two is used alone, which is confusing. Llama-3.1-70B shows somewhat high ASRs on ciphers that have a 0% decryption rate, which is also confusing.

It would help to aggregate a collection of known and novel ciphers with similar decryption rates and show a difference in the average ASR of these collections. This would support the argument that novel ciphers are open vulnerability, and it could also help identify which models have been exposed to safety training regarding known ciphers. For the argument that improved reasoning capabilities are correlated with increased cipher ASR, aggregating data over more models would be helpful. If the models were similar in architecture or training process it would add strength to the argument.

For the idea of layering ciphers with LACE. the paper shows some evidence that this increases ASR compared to a single cipher with a similar decryption difficulty. Further evidence for this would be very interesting, and it would be especially interesting if known ciphers with low ASR can be layered to achieve a high ASR.